# How social media data are being used to research the experience of mourning: A scoping review

**Julia Muller Spiti**[1]\*, **Ellen Davies**[1,2], **Paul McLiesh**[1], **Janet Kelly**[1]

**1** Adelaide Nursing School, Faculty of Health and Medical Sciences, University of Adelaide, Adelaide, Australia, **2** Adelaide Health Simulation, Faculty of Health and Medical Sciences, University of Adelaide, Adelaide, Australia

\* Julia.MullerSpiti@adelaide.edu.au

## Abstract

### Background

Increasingly, people are using social media (SM) to express grief, and researchers are using this data to investigate the phenomenon of mourning. As this research progresses, it is important to understand how studies are being conducted and how authors are approaching ethical challenges related to SM data.

### Objective

The aim of this review was to explore how SM data are being used to research experiences of mourning through the following questions: a) 'Which topics related to mourning are being studied?'; b) 'What study designs have been used to analyse SM data'; c) 'What type of data (natural or generated) have been used?'; and d) 'How are ethical decisions being considered?'.

### Methods

The JBI Scoping Review methodology guided this review. Eligibility criteria were determined using the PCC framework, and relevant key words and phrases derived from these criteria were used to search eight databases in September 2021 (CINAHL, Embase, LILACS, OpenGrey, ProQuest, PsycINFO, PubMed and Scopus). The Preferred Reporting Items for Systematic Reviews and Meta-Analyses Extension for Scoping Reviews (PRISMA-ScR) guidelines were used to report the results of this review.

### Results

Database searches resulted in 3418 records, of which, 89 met eligibility criteria. Four categories of grief and mourning were identified. Most records were qualitative in nature and used natural data. Only 20% of records reported ethics approval by an Institutional Review Board, with several including measures to protect participants, for example, using pseudonyms.

**Data Availability Statement:** All relevant data are within the manuscript and its Supporting Information files.

**Funding:** The authors received no specific funding for this work.

**Competing interests:** The authors have declared that no competing interests exist.

**Abbreviations:** AoIR, association of internet researchers; IRB, institutional review board; NHMRC, National Health and Medical Research Council; SM, social media; UGC, user generated content.

## Conclusions

This unique review mapped the diverse range of mourning-related topics that have been investigated using SM data and highlighted the variability in approaches to data analysis. Ethical concerns relating to SM data collection are identified and discussed. This is an emerging and rapidly changing field of research that offers new opportunities and challenges for exploring the phenomenon of mourning.

## Introduction

Social Media (SM) platforms have altered, arguably forever, the way humans communicate and express themselves. As of January 2021, 4.6 billion of the 7.8 billion people in the world had access to the internet, and 4.2 billion were active SM users [1]. Unsurprisingly, the dynamic nature of large volumes of user-generated content (UGC), and high level of self-disclosure that is available on SM platforms has drawn the attention of researchers. In 2004, Donath and Boyd [2] pioneered SM research, discussing public displays of connection and how the online environment is used as a space for self-representation. Clarke and Van Amerom [3] were among the first authors to utilize data from SM to gain an understanding of social phenomena. Subsequently, several disciplines have used data from SM to inform decisions and understand social trends, including marketing [4], journalism [5] and health sciences [6]. The access that researchers have to large amounts of data, paired with the level of disclosure that is demonstrated on SM platforms, offers the opportunity to investigate social phenomena in a way that has not previously been possible with traditional methods of research [7].

The definition of SM has evolved over the last 10 years as society has adopted new versions of technology [8]. McCay-Peet and Quan-Haase [9] have proposed that "social media are web-based services that allow individuals, communities, and organizations to collaborate, connect, interact, and build community by enabling them to create, co-create, modify, share, and engage with user-generated content that is easily accessible" [9 p.17]. While SM platforms may differ in purpose, they are essentially all internet-based forms for communicating UGC [9].

Two perceived benefits of SM are the invisibility and anonymity it offers to users. The largely text-driven environments can eliminate concerns about physical appearance, tone of voice and body language when sharing messages, and delays or eliminates experiencing any reaction or feedback from recipients [10]. If users want to take a step beyond being physically invisible, there are options to remain anonymous when posting. Anonymity online is the act of hiding one's true self from others, and is known to provide users with a sense that their actions on SM platforms will have no impact on their 'real' (offline) lives [10].

Invisibility and anonymity are not the only benefits recognized by people who share personal information or opinions online. Researchers have found that disclosing stressful or sensitive information on SM has benefited individuals, by allowing them to connect with people with whom they identify [11, 12]. When analyzing posts and pictures linked to the hashtag "Depression" on Instagram, Andalibi [13] found clear evidence of social support and a sense of community. This was confirmed by Zhang [14] when investigating the influence of SM on university students' mental health. Zhang found that self-disclosure on SM was higher during stressful life events and was positively associated with life satisfaction and reduced incidence of depression. People are more likely to disclose thoughts and emotions on SM because of

anonymity and invisibility, but they also use SM for self-disclosure because it has the potential to improve their well-being through social connection [15].

Data from SM can come in different forms such as text, images, or videos. Different SM platforms allow for different forms of expression and target specific demographics. For example, YouTube is a video sharing platform [16] and most users are males aged 18 to 34 [17]. Whereas Snapchat, a messaging and photo sharing platform, has mostly female users aged from 13 to 24 [18]. Social media are unique in the sense that large amounts of data are available from people from varied walks of life and demographics [1]. The access to data from diverse demographics allows researchers to explore specific topics and gain greater (or at least different) understanding of peoples' experiences [19–21].

There are two main types of data used in SM research: natural, and generated data. Observational SM research relies on natural data, which refers to data collected without the awareness of participants [22]. An example of this type of data collection was published by Hilton [23], who analyzed posts from Twitter to investigate self-harm, but had no influence over the generation of data. The use of natural data has been effective in deepening researchers' understanding of sensitive topics, such as miscarriage [21], eating disorders [24], and even to identify shifts from mental health discourse to suicidal ideation [25].

Interactive SM research uses data generated through the active involvement of the researcher on SM platforms [26]. This may be initiated by the researcher extending a 'friendship request' to a prospective participant or through following someone on Twitter to gain access to posts. The researcher may also contact potential participants with requests to create content, or may already be active on the platform from which data will be extracted [7]. This approach has been successfully employed by Caplan [27], who analyzed personal accounts of poverty posted on Reddit in response to an anonymous question from the researcher.

While research using SM data can be valuable in the quest to understand social phenomena, it raises significant questions regarding privacy and ethical conduct in research. These ethical considerations have been explored and expanded upon by several authors. Elgesem [28], in 2002, published a seminal discussion paper exploring questions about consent and the private vs public nature of the data. Also in 2002, the first set of recommendations for ethical conduct in internet research were published by the Association of Internet researchers [29], and were further updated in 2012 [30] and 2020 [31]. De Montjoye et al. [32, 33] highlighted the challenges that modern information technologies bring to individuals' privacy. The issues discussed in these documents are as relevant today–if not more–as they were 20 years ago. Questions that arise include: who 'owns' data from SM–do they fall into the private or public domain? Are SM users aware their posts may be used in research, and would they consent if they were? Should the original intent of the poster be respected? These are pertinent questions that have been raised by the research community [26, 34, 35], as well as by SM users [36] and are of particular importance when data are used to investigate vulnerable populations, such as the bereaved, or are related to practices that have been culturally considered intimate, such as mourning a loss.

The use of SM data to explore grief and mourning is the focus of this review. Grief, in the context of this review, is defined as the intense emotion, sorrow or regret keenly felt following a loss, whereas mourning refers to the practices performed by people in response to their grief [37–39]. Bearing witness to death is a natural part of the life experience, but for many, this experience is radically different to bygone eras because of SM. Peoples' experiences of death now invade our daily lives via televisions, radios, portable devices, and mobile phones [40]. It is almost impossible to be ignorant of a celebrity's death, or the occurrence of a natural disaster on the other side of the world [41].

In pre-modern societies, death resulted in a *bereaved community* [41]. Families lived together or in geographical proximity–neighbors knew and depended on each other for survival. When someone in a community died, all members of that community experienced loss, and would mourn together through rituals designed to memorialize the deceased. Modern societies, on the other hand, are said to have produced *bereaved individuals* [41]. Urban developments accompanied by geographical mobility have resulted in a reduced sense of community, leading to increasingly private and isolated experiences of grief [42].

Walter et al. [41] suggest that in the post-modern society, with the advent of the internet, we are offered the opportunity to grieve as a community once again, resulting in *communities of the bereaved.* This is possible because the internet can connect those who have suffered similar loss. Online communities provide a space for connection and public expression of grief and as such, represent a profound change in how people mourn when compared to the pre-internet era [41].

Expressing grief online has become so commonplace that a new term has been coined to represent this behavior: 'Mourning 2.0'. This term alludes to Web 2.0 –the web of interaction and sharing of information, as opposed to Web 1.0 where information was available without interaction. It encapsulates how mourning has expanded from the private sphere to the public arena [43]. There are numerous support groups available for grievers on SM. The social support offered to individuals on SM contributes to the recognition of their grief, through the acknowledgement of their loss and validation of their feelings [41].

While SM research about the experience of mourning has aided in understanding the post-modern expression of grief, particularly in generations Y and Z (70% of SM users [1]), no study has been conducted to provide a comprehensive overview of the topics, study designs, type of data and ethical considerations involved in SM research about mourning. As such, the overarching aim of this review is to explore how SM data are being used to research the experience of mourning.

There is value in mapping how SM data are being used in research because the internet has changed the way we mourn, with an increasing number of people not only turning to SM to express their grief, but also potentially disclosing more information than they would in face-to-face interactions. Researchers have identified the opportunity to capture and understand the experience of mourning in a different way by using SM data [44–46]. However, as the research output about mourning online increases, it is necessary to understand how these studies are being conducted for two main reasons: to inform future research, particularly in vulnerable populations, and to report and discuss the ethical challenges inherent to the use of natural data from SM platforms.

## Methodology

Scoping reviews are an increasingly popular approach to reviewing the literature to comprehensively summarize and synthesize knowledge [47–49]. Scoping reviews address broad research questions, are exploratory and descriptive in nature [49], and are usually conducted to explore the breadth and depth of the literature on a particular topic, to map and summarise evidence, and inform the direction of future research [50].

Scoping reviews are indicated for a variety of reasons, for example as a precursor to a systematic review; to identify the types of available evidence or how research is conducted in a given field; to identify and analyze knowledge gaps; or to clarify key concepts in the literature [51]. Whilst scoping reviews are successfully used to explore established fields of research, this type of review is particularly useful in emerging areas of research where there is variability in methodologies and approaches to data collection and analysis, as well as poor indexing, and a

distribution of research across different academic disciplines [52–55]. The decision flowchart available in Pollock et al. [56] was used to guide this decision to adopt a scoping review approach to this review and the selection of the JBI methodology for scoping reviews, as it is currently the most detailed and rigorous approach available [50].

## Review questions

As this is an emerging field of research, with studies being undertaken in diverse disciplines and with a variety of research methodologies and approaches to data analysis, the primary question for this review was 'How are social media data being used to research the experience of mourning?'. This question was intentionally broad, to capture the extent and breadth of literature relating to the central topic.

Four specific sub questions were also considered and include the following: a) 'Which topics related to mourning are being studied using SM data?'; b) 'What study designs have been employed in the analysis of SM data about the experience of mourning?'; c) 'What type of data (natural or generated) have been predominantly used in SM research about the experience of mourning?'; and d) 'How are ethical aspects considered in the published research?'. These were constructed to provide focus for the exploration of the included studies and to provide guidance for data extraction and analysis.

## Protocol and registration

A protocol was developed in accordance with the Scoping Review methodology proposed by Arksey and O'Malley [48] and JBI [50], and outlined eligibility criteria, search strategy, study selection and data extraction for this review. The final version of the protocol was registered prospectively with Open Science Framework (https://osf.io/a2udy/). The reporting of this review is guided by the PRISMA Extension for Scoping Reviews reporting guidelines [57] (S1 Appendix).

## Eligibility criteria

Eligibility criteria for this review are described using the Participants, Concept and Context (PCC) framework [50]. Participants included people that had expressed grief on SM—such as posting messages to the deceased on SM or creating online memorials to celebrate the deceased's life. There were no restrictions based on age or other demographic aspects. The concept explored in this review included records that report primary research projects that analyzed SM data to explore the experience of mourning. Records that report the use of SM to recruit participants but did not collect data from SM were not eligible for inclusion. All forms of online mourning were considered, including written, audio-visual, and photographic expression. The context included records from any academic discipline where data was collected from SM regardless of geographical location or type of SM platform.

Records of published and unpublished primary research studies, published in either English or Portuguese, were eligible for inclusion in this review. There were no limitations relating to study design or approach to data analysis. There were no limitations on year of publication as the analysis of data from SM for research purposes is a relatively recent phenomenon and is therefore chronologically self-limited.

## Search strategy and information sources

On the advice of the academic librarian, an initial limited search of MEDLINE and CINAHL was undertaken to identify eligible records. The text words contained in the titles and abstracts

of relevant records, and the index terms used to describe the records were used to develop a full search strategy for CINAHL (see S2 Appendix). The search strategy, including all identified keywords and index terms, was adapted for each included information source. This process was guided by the assistance of an academic librarian. The reference lists of all records retrieved for full text review were screened for additional papers. The search was completed in September 2021. The databases that were searched included CINAHL (EBSCO), Embase (Elsevier), LILACS (BIREME), OpenGrey (INIST-CNRS), ProQuest Dissertations and Theses Global (ProQuest), PsycINFO (APA), PubMed (NCBI), and Scopus (Elsevier).

## Selection of sources of evidence

All identified citations were collated and uploaded into the reference management system EndNote (Clarivate Analytics, PA, USA—Version X9) and duplicates were removed. In preparation for title and abstract screening the reviewers met several times to discuss nuanced elements of the inclusion criteria, in this emerging area of research, and to pilot the screening. This was an iterative process that provided clarity for the three reviewers prior to the lead reviewer (JMS) proceeding with the title and abstract screening.

The full text versions of selected records were screened independently by two reviewers (JMS, JK and PM). Reasons for exclusion of full text records were recorded and are presented in the PRISMA flow chart (Fig 1). JBI Portugal and Brazil were contacted for assistance with screening records published in Portuguese. A reviewer from JBI Portugal assisted with screening full text records published in Portuguese, which resulted in the inclusion of 5 records. No disagreements arose between reviewers at any stage of the study selection process. The results of the search are presented in the PRISMA flow diagram as per the PRISMA 2020 guidelines [58].

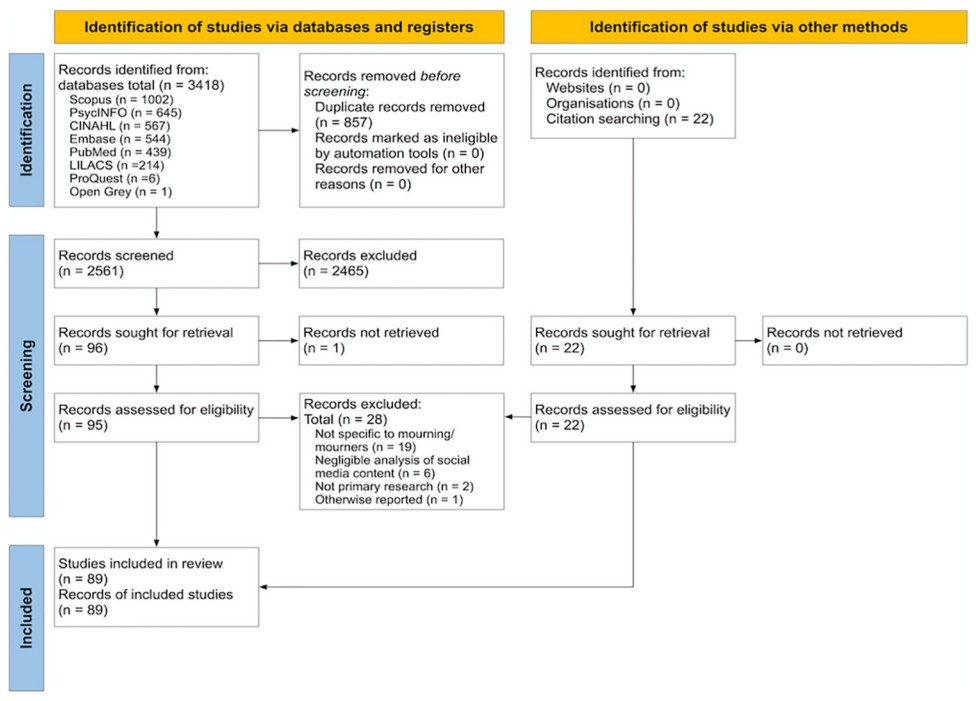

**Fig 1. PRISMA flowchart.**

## Data charting process

Data were extracted from included records by the lead reviewer (JMS) after pilot-testing of the data extraction tool by three reviewers (JMS, JK & PM). The extraction was piloted on two occasions using randomly selected samples of five documents. Extractions were compared and found to be congruent. The lead reviewer proceeded with the extraction of the remaining records. As per JBI guidance for conducting SC Re the reviewers met on several occasions to discuss the data that was being extracted to ensure the data being extracted were sufficient to address the review questions. These data included specific details about the topic investigated, study design, type of data, and ethical considerations.

## Results

The database search resulted in 3418 records (see PRISMA diagram, Fig 1). After duplicate removal and title and abstract screening, 95 full text records were assessed for eligibility. Of these, sixteen were excluded as they did not focus on mourning; six were excluded as the analysis of SM content was negligible and two were excluded as they did not report primary research projects. An additional 22 records that met the eligibility criteria were identified through pearling the included records. Of these,

three were excluded as they did not focus on mourning. One record was a journal article that presented the methodology and results from a PhD thesis: therefore, the article was excluded, and the thesis remained, as it described each aspect of the study more comprehensively. In total, 89 records met the eligibility criteria (full citations listed and complete data from the included records are available in S3 Appendix). The search was completed in September 2021.

Most records included in this review were conducted by researchers based in North America (n = 51; 57%), Europe (n = 22; 25%) [20, 46, 59–77], and Oceania (n = 8; 9%) [78–84] (Table 1). All included records were published between 2000 and 2021, and each record reported on a distinct study. Journal articles represented most of the sample (n = 76; 85%), followed by theses (n = 9; 10%) [68, 69, 85–91], and conference proceedings (n = 4; 5%) [92–95]. The most prevalent academic disciplines of first authors included Communication (n = 27; 30%) [45, 46, 59, 61, 62, 87, 88, 91, 95–113]. Psychology (n = 14; 16%) [20, 67, 74, 75, 79–81, 89, 114–119], Nursing (n = 7; 8%) [120–125], Sociology (n = 7; 8%) [43, 44, 66, 77, 78, 126, 127], and others (n = 34; 38%). The most widely used type of SM was social networking sites (n = 35; 39%), with Facebook the main platform used (n = 29; 32%) (Fig 2).

Data were collected in written form, as well as images and audio-visual content. Forty-eight records (54%) did not report how content was collected, whereas manual collection was reported in 33 (37%) records [19, 44, 46, 60, 61, 69, 73, 79, 86, 89–91, 98, 103, 104, 107, 108, 111, 112, 115, 120, 125, 126, 128–132], and automated data collection was used in 6 (7%) records [65, 95, 116, 133]. Two records [68, 70] reported using both strategies to collect data. Most records used written units for analysis (n = 71; 87%). In these records, there was a large variation in sample size (range = 8–291443 units). Two records used images [127] or audio-visual posts [78], one record analyzed emojis from posts [84], and 9 records analyzed multiple data types [20, 46, 63, 68, 83, 85, 88, 109, 126].

## Study designs and topics that were explored

While most records did not specify the overarching methodology underpinning their research (n = 67; 75%), of the records that did mention a methodology, two main methodologies were used by researchers to explore mourning on SM: ethnography (n = 13; 15%) [61, 72, 73, 83, 88, 90, 109, 111, 126], including digital ethnography–also referred to as netnography, or virtual, or

**Table 1. Summary of 89 included records.**

| Records characteristics | Included records, n (%) | Records characteristics | Included records, n (%) |
|---|---|---|---|
| **Region of first author** | | **Method for data collection** | |
| North America | 51 (57) | Unclear | 48 (54) |
| Europe | 22 (25) | Manual | 33 (37) |
| Oceania | 8 (9) | Automated | 6 (7) |
| South America | 4 (5) | Combination of manual and automated | 2 (2) |
| Asia | 3 (3) | **Type of data** | |
| Middle East | 1 (1) | Natural data | 83 (94) |
| **Publication Type** | | Generated data | 4 (4) |
| Journal Article | 76 (85) | Other | 2 (2) |
| Conference proceedings | 4 (5) | **Approach to data analysis** | |
| Dissertation/thesis | 9 (10) | Qualitative | 70 (79) |
| **Discipline of first author** | | *Content Analysis* | *21 (24)* |
| Communication | 27 (30) | *Coding* | *12 (14)* |
| Psychology | 14 (16) | *Thematic Analysis* | *11 (12)* |
| Nursing | 7 (8) | *Textual Analysis* | *6 (7)* |
| Sociology | 7 (8) | *Other* | *20 (22)* |
| Social Work | 6 (7) | Mixed Methods | 10 (11) |
| Education | 4 (5) | Quantitative | 9 (10) |
| Other | 24 (26) | *Sentiment Analysis* | *8 (9)* |
| **Sample type** | | *Content Analysis* | *1 (1)* |
| Written text only | 76 (86) | **Ethics approval** | |
| Video only | 2 (2) | No | 71 (80) |
| Image only | 1 (1) | Yes | 18 (20) |
| Emojis | 1 (1) | **Exemption granted by relevant ethics committee** | 3 (4) |
| Multiple sample types | 9 (10) | | |
| **Sample size, median (range)** | | | |
| Posts | 588 (8–291443) | | |
| Images | 361 (229–493) | | |
| Videos | 31 (1–126) | | |

online ethnography, and grounded theory (n = 7; 8%) [44, 64, 66, 87, 106, 108, 117]. Critical realism [71] and interpretive phenomenology analysis [89] were also mentioned in one record each.

Records that mentioned grounded theory as the underpinning methodology have described data analysis using content, thematic or discourse analysis. And records that mentioned ethnography have described data analysis using content, textual, narrative, and critical discourse analysis. Out of the 13 records that have used ethnography, nine reported the use of a digital form of ethnography, with four of them mentioning Netnography [61, 72, 73, 130]—a term coined by Kozinets [134] who developed an adaptation of traditional ethnography to suit the context of online communities, specifically in marketing research. However, it is unclear whether these records have followed the principles of Netnography as described by Kozinets. Even though most records did not adequately or appropriately report the methodological approach being used in their research, the analysis of SM data relating to mourning was described in varying degrees of detail and approached in a variety of ways.

A total of 20 different approaches to data analysis were identified and represented in Tables 2–5. Of the 89 records, 70 (79%) were qualitative, 10 were mixed methods (11%), and nine were quantitative records (10%). Those that employed a qualitative research approach used

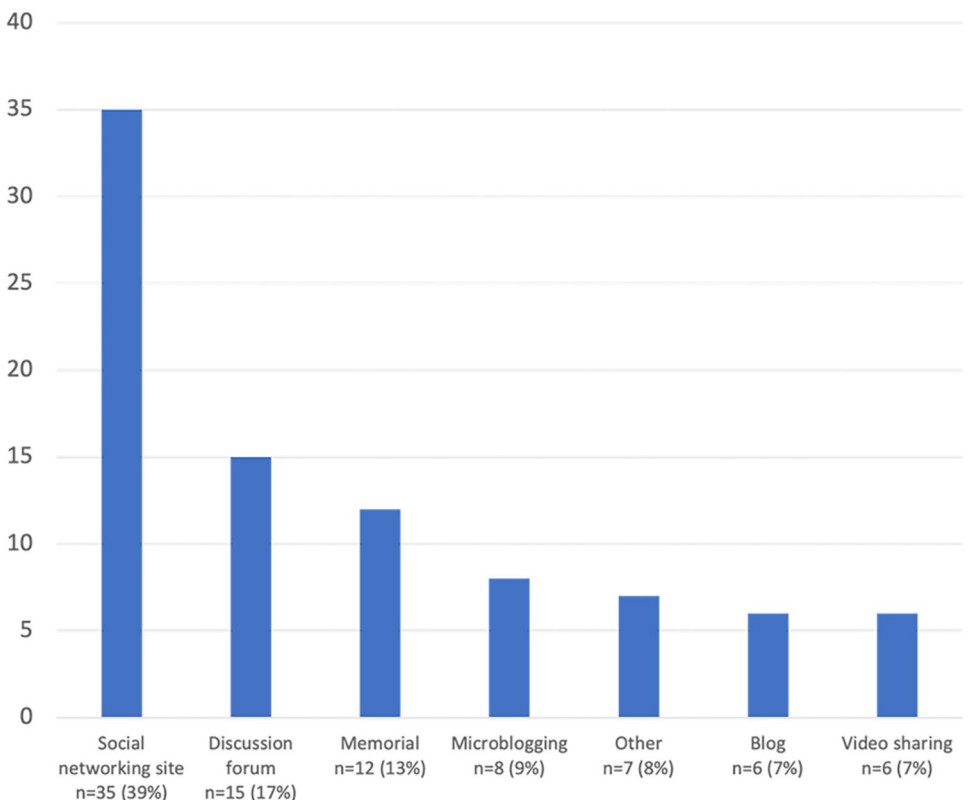

**Fig 2. Social media platforms used for data collection.**

primarily content analysis (n = 21; 24%), coding (n = 12; 14%), and thematic analysis (n = 11; 12%). The most frequently combined approaches to data analysis in the mixed methods records were thematic analysis and descriptive statistical analysis (n = 3; 4%) to interpret and represent the data. In the quantitative records data were analyzed predominantly using sentiment analysis [70, 75, 84, 93, 94, 114–116]. A large variety of topics were investigated and reported in the included records. These have been identified and divided into four categories to facilitate the representation of data in a meaningful way (Fig 3).

Categories and the allocation of topics were determined by how the authors described the expressions of grief and the reviewers' perception of the data. A table for each category was developed to map the study designs used for data analysis, the year of publication, the country of the first author, the type of data used, the type of SM platform where data were collected, as well as ethics approval by Institutional Review Board (Tables 2–5).

## Type of data collected and analyzed

Most included records used natural data in their analysis (n = 83; 94%). Four records (4%) [83, 85, 90, 125] used generated data, where authors were, or had been, active in the setting where the data was collected. In one (1%) [120] record, it was not possible to determine if the researcher had any influence on the generation of data, and one (1%) [135] record reported the use of both natural data and generated data.

Most mixed methods and quantitative records used natural data. The four records that used generated data, as well as the record where the data source was ambiguous, were qualitative in nature.

**Table 2. Death of a loved one.**

| Approach to data analysis | | Death of a loved one n = 32 (36%) — Unspecified loved one n = 8 (9%) | Own child n = 8 (9%) | Friend n = 7 (8%) | Miscarriage, perinatal loss, and stillbirth n = 5 (6%) | Other n = 4 (4%) |
|---|---|---|---|---|---|---|
| Qualitative | Content Analysis | Wittenberg-Lyles et al. 2015 (USA) SNS, GD | Aho, Paavilainen & Kaunonen 2012 (USA) F, U | Bouc, Han & Pennington 2016 (USA) NSN, ND | Bakker & Paris 2013 (USA) F, ND | |
| | | Keskinen, Kaunonen & Aho 2019 (Finland) SNS, ND | Musambira, Hastings & Hoover 2006 (USA) F, ND | Bousso et al. 2014 (Brazil)[†] SNS, ND | Sawicka 2017 (Poland) O, ND | |
| | | Selman et al. 2021 (UK)[†] MB, ND | | | | |
| | Narrative Analysis | | Frizzo, Bousso, De Faria & De Sa 2017 (Brazil)[†] B, ND | Giaxoglou 2015 (UK) SNS, ND | | **Spouse n = 2 (3%)** |
| | | | | | | Frizzo et al. 2017 (Brazil)[†] SNS, ND |
| | | | | | | McDonald-Kenworthy 2012 (USA) F, GD |
| | Coding | | | Cesare & Branstad 2018 (USA) MB, ND | | |
| | Thematic Analysis | | | | Hayman, Chamberlain & Hopner 2018 (New Zealand)[†] SNS, ND | |
| | Textual Analysis | | Finlay & Krueger 2011 (USA) M, ND | | | |
| | Discourse Analysis | DeGroot 2009 (USA) SNS, ND | | | | |
| | | Pawelczyk 2013 (Poland) M, ND | | | | |
| | Rhetorical Critical Approach | | DeGroot & Carmack 2013 (USA) B, ND | | | |
| | Contrapuntal Analysis | | | | | **Sibling n = 1 (1%)** |
| | | | | | | Halliwell & Franken 2016 (USA) F, ND |
| | Generative Rhetorical Analysis | | | Brooks 2014 (USA) SNS, GD | | |
| | Unclear | | Christensen et al. 2017 (Denmark) O, ND | | | |
| Quantitative | Sentiment Analysis | Han et al. 2021 (China)[†] MB, ND | Liu et al. 2019 (China)[†] MB, ND | Brubaker, Kivran-Swaine, Taber & Hayes 2012 (USA) SNS, ND | | |
| | Content Analysis | Doveling 2015 (Germany) SNS, ND | | | | |
| Mixed Methods | Thematic Analysis and Descriptive Statistics | | Hastings, Musambira & Hoover 2007 (USA) F, ND | Brubaker & Hayes 2011 (USA) SNS, ND | | |
| | Textual Analysis and Sentiment Analysis | Gray 2019 (USA) SNS, ND | | | | |
| | Thematic Analysis and Latent Dirichlet Allocation | | | | Cesare et al. 2020 (USA) MB, ND | |
| | Coding and Descriptive Statistics | | | | | **Mother n = 1 (1%)** |
| | | | | | | Nager & de Vries 2004 (USA) M, ND |
| | Qualitative unclear and Descriptive Statistics | | | | Sani, Dimanche & Bacque 2019 (France) VS, ND | |

Notes

[†] Article stated ethics approval by Institutional Review Board. Type of platform: B, Blog; F, Forum; M, Memorial; MB, Microblogging; O, Other; SNS, Social network site; VS, Video sharing. Type of data: ND (Natural data); GD (Generated data), U, (Unclear).

**Table 3. Grief.**

| Approach to data analysis | | Disenfranchised Grief n = 15 (17%) | | | The experience or grief itself n = 8 (9%) |
|---|---|---|---|---|---|
| | | Suicide n = 9 (10%) | Death of pet n = 3 (3%) | Other n = 3 (3%) | |
| Qualitative | Content Analysis | Schotanus-Dijkstra et al. 2014 (the Netherlands) F, ND | | | Children, adolescents' and young adult's grief n = 2 (3%) |
| | | | | | Doveling 2015 (Germany) SNS, ND |
| | | | | | Peruzzo 2007 (Brazil)[†] SNS, ND |
| | | | | | Communicating Grief n = 1 (1%) |
| | | | | | Eriksson Krutrok 2021 (Sweden) MB, ND |
| | Coding | Pritchard & Buckle 2018 (Canada)[†] F, ND | | | Athlete's concussion n = 1 (1%) |
| | | | | | Cassilo & Sanderson 2019 (USA) B, ND |
| | | | | | Death of adolescent peer n = 1 (1%) |
| | | | | | Williams & Merten 2009 (USA) SNS, ND |
| | Thematic Analysis | Krysinska & Andriessen 2015 (Australia) M, ND | Vitak et al. 2017 (USA)[†] SNS, ND | Traumatic birth n = 1 (1%) | Being a mother with cancer n = 1 (1%) |
| | | Krysinska, Andriessen & Corveleyn 2014 (Australia) M, ND | Laing & Maylea 2018 (Australia) B, ND | DeGroot & Vik 2017 (USA) SNS, ND | Croson & Keim-Malpass 2016 (USA) B, ND |
| | | Scott 2012 (UK)[†] M, ND | | | |
| | Textual Analysis | | | Abortion n = 1 (1%) | Death of employer n = 1 (1%) |
| | | | | Heathcote 2014 (Australia)[†] O, GD | Babis 2020 (Israel) SNS, ND |
| | Unclear | Hagstrom 2017 (Sweden) F, ND | | | |
| | | Hagstrom 2017 (Sweden) F, ND | | | |
| Quantitative | Sentiment Analysis | Lester 2012 (USA) M, ND | Lyons et al. 2020 (UK) F, ND | | Communicating Grief n = 1 (1%) |
| | | Scourfield et al. 2019 (UK) SNS, ND | | | Getty et al. 2011 (Canada) SNS, ND |
| Mixed Methods | Thematic Analysis and Descriptive Statistics | | | AIDS n = 1 (1%) | |
| | | | | Blando, Graves-Ferrick & Goecke 2004 (USA) M, ND | |

Notes

[†] Article stated ethics approval by Institutional Review Board. Type of platform: B, Blog; F, Forum; M, Memorial; MB, Microblogging; O, Other; SNS, Social network site; VS, Video sharing. Type of data: ND (Natural data); GD, (Generated data), U, (Unclear).

## General aims of included records

Even though the aims of included records were not encompassed in our research questions, during data extraction, it became clear that gaining insight into the overall purpose of studies

**Table 4. Unspecified death.**

| Approach to data analysis | | Unspecified death n = 18 (20%) |
|---|---|---|
| **Qualitative** | **Content Analysis** | Carroll & Landry 2010 (USA) SNS, ND |
| | | De Vries & Rutherford 2004 (USA) M, ND |
| | | Forman, Kerr & Gil-Egui 2012 (USA) SNS, ND |
| | | Irwin 2015 (USA) SNS, ND |
| | | Hasting, Hoover & Musambira 2005 (USA) F, ND |
| | **Coding** | DeGroot 2014 (USA) SNS, ND |
| | | Dinning-Brinkmann 2010 (USA) VS, ND |
| | | Giaxoglou 2014 (UK) SNS, ND |
| | | Gibson 2016 (Australia) VS, ND |
| | | Karkar & Burke 2020 (Australia)[†] B, ND |
| | | Roberts & Vidal 2000 (USA) M, ND |
| | **Textual Analysis** | Keye 2017 (USA) SNS, ND |
| | | Willis & Ferrucci 2017 (USA) SNS, ND |
| | **Discourse Analysis** | Paulus & Varga 2015 (USA)[†] F, ND |
| | | Varga & Paulus 2014 (USA) F, ND |
| | **Qualitative Document Analysis** | Kasket 2012 (UK)[†] SNS, ND |
| | **Unclear** | Huberman 2017 (USA) M, ND |
| **Quantitative** | **Sentiment Analysis** | Xu, Manrique & Pereira Nunes 2021 (Australia) MB, ND |

Notes

[†] Article stated ethics approval by Institutional Review Board. Type of platform: B, Blog; F, Forum; M, Memorial; MB, Microblogging; O, Other; SNS, Social network site; VS, Video sharing. Type of data: ND, (Natural data); GD, (Generated data), U, (Unclear).

would add depth to the results of this review. Authors have described that their objectives involved the investigation of the ongoing engagement of survivors with the online presence of the deceased [74, 87, 92, 93, 102, 136, 137], the understanding of how people use SM to make sense of death [87, 90, 112, 136], the role of virtual interaction in mourning [46, 62, 109, 110], as well as the role of social support in online mourning [79]. Researchers have also been motivated by the opportunity to gain insight into people's reasons for mourning online [129], as well as the phenomenon of mourning among strangers [61, 78]. While the findings of the records are not the focus of this review, understanding why researchers conducted their studies contributes to the mapping of the use of SM data in research about the experience of mourning.

Two records, both from 2021, reported on research conducted using data from SM related to COVID-19. Han et al. [114] explored the impact of the COVID-19 pandemic on the bereaved, finding that the bereaved due to COVID-19 were more preoccupied with their grief, but displayed lower depression scores, compared to non-COVID-19 bereaved individuals. Selman et al. [71] explored the views and experiences of SM users resulting from knowing that someone they care about died without a family member or friend present and discussed the specific sadness of not being able to say goodbye. Both records collected data from microblogging platforms and obtained ethics approval.

## Reporting of ethical considerations

Ethics approval from an Institutional Review Board (IRB) was not reported in the majority of records (n = 71; 80%), with many justifying this by stating that the data is considered public [19, 43, 45, 46, 60, 63, 86, 88, 104, 108, 112]. The authors of 21 studies applied for ethics

Table 5. Mediatized death.

| Mediatized Death<br>n = 16 (18%)<br><br>Approach to data analysis | | Famous people<br>n = 11 (12%) | Non-famous people<br>n = 5 (6%) |
|---|---|---|---|
| Qualitative | Content Analysis | Bingaman 2020 (USA)<br>SNS, ND | Klastrup 2015 (Denmark)<br>SNS, ND |
| | | | Pearce 2020 (USA)<br>SNS, ND |
| | Coding | Klastrup 2018 (Denmark)<br>SNS, ND | |
| | | Radford & Bloch 2012 (Canada)<br>F, ND | |
| | Thematic Analysis | Akhter & Tetteh 2021 (USA)<br>MB, ND | |
| | | DeGroot & Leith 2018 (USA)<br>SNS, ND | |
| | | Sanderson & Cheong 2010 (USA)<br>MULT, ND | |
| | Textual Analysis | Campbell & Smith 2015 (USA)<br>M, ND | |
| | Discourse Analysis | Pattwell 2017 (USA)<br>O, ND | Scott 2017 (UK)<br>VS, ND |
| | Critical Discourse Analysis | Harju 2015 (Finland)<br>VS, ND | |
| | Unclear | | Foot, Warnick & Schneider 2005 (USA)<br>O, ND |
| Mixed Methods | Textual Analysis and Natural Language Processing | | Patton et al. 2018 (USA)<br>MB, ND |
| | Coding and Sentiment Analysis | Stone & Pennebaker 2002 (USA)<br>O, ND | |
| | Coding and Descriptive Statistics | Alemi, Pazoki & Rezanejad 2021 (Iran)<br>SNS, ND & GD | |

Notes: [†] Article stated ethics approval by Institutional Review Board. Type of platform: B, Blog; F, Forum; M, Memorial; MB, Microblogging; O, Other; SNS, Social network site; VS, Video sharing. Type of data: ND, (Natural data); GD, (Generated data), U, (Unclear).

approval; 18 had their applications approved and three were provided with an exemption by their IRB with the justification that SM data are considered public domain and therefore consent is not required [19, 124]. In disciplines related to health sciences 60% of records reported approval by an IRB, and 17% from disciplines related to social sciences.

Of the 18 (20%) [69, 71, 74, 77, 79, 83, 87, 90, 95, 114, 117, 119, 121, 131] records that obtained ethics approval, 16 were published after 2012, the year that the Association of Internet Researchers (AoIR) published updated recommendations for ethical conduct of online research [30]. Sixteen of these 18 records were qualitative in nature, and two were quantitative [114, 116]. None of the mixed methods records reported obtaining ethics approval. Informed consent from participants was reported in six records, four that used natural data [69, 71, 119, 122], one that used generated data [90], and one where the type of data was unclear [120]. The AoIR guidelines are mentioned in 5 (6%) of the records that obtained ethics approval [19, 69, 77, 121, 131].

Regarding the protection of the identity of SM posters, 36 (40%) records described measures to protect the anonymity of posters [20, 21, 44, 59, 62, 66, 67, 71–74, 76, 77, 80–82, 89–91, 95, 96, 101, 114, 116, 117, 119, 121, 123, 126, 129–131, 133, 138–140], by changing their

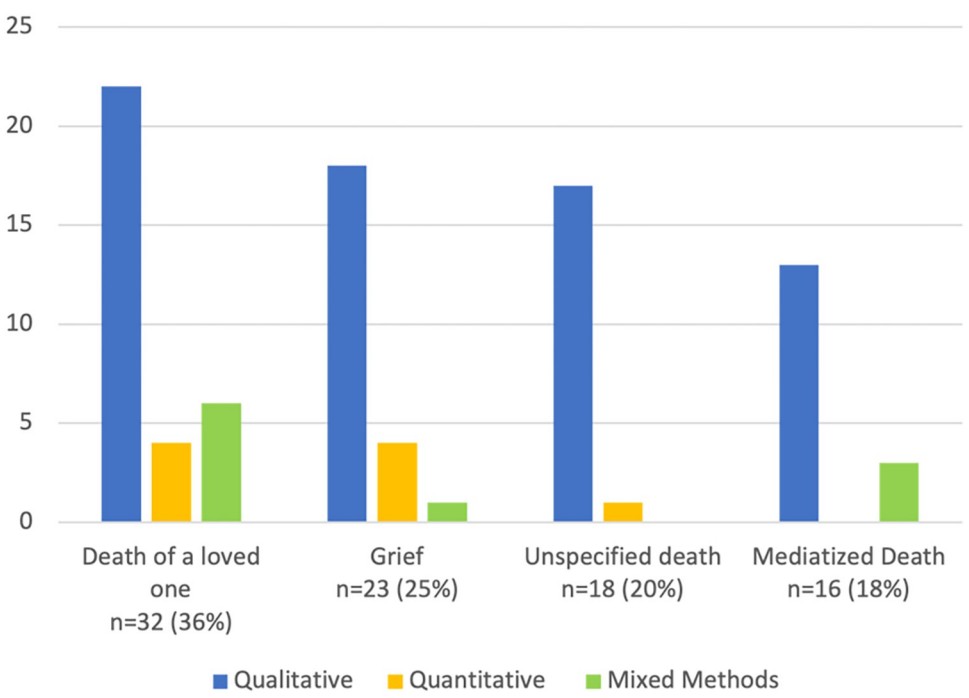

**Fig 3. Research approach by category.**

profile names to pseudonyms, and/or not publishing any identifying information to protect posters' privacy. However, 50 (56%) records did not include any statement outlining considerations regarding users' privacy or other ethical considerations.

## Discussion

This scoping review explored how SM data are used to research the experience of mourning. Specifically, it has identified the topics related to mourning being explored using SM data, the study designs employed by researchers, the type of data used to research the experience of mourning (natural data or generated), and the ethical aspects considered in published research.

After an extensive search of the literature, 89 records were included in this review. From these records, it is evident that the use of SM data has become an increasingly popular avenue to explore the phenomenon of mourning. A wide range of topics have been researched using data from SM platforms. The loss of a loved one was a frequently investigated topic, representing over 30% of the included records. Another significant area of investigation was grief: the experience of grief itself and disenfranchised grief.

When mourners have their right to grieve denied, or the legitimacy of their grief questioned, be it for reasons related to the way someone grieves, the nature of the loss or the nature of the relationship, their grief is referred to as disenfranchised grief [141]. In this review, it was found that SM platforms offered users who experience disenfranchised grief the opportunity to form community. This was evident in people who have lost a loved one to suicide [67, 69, 70, 72, 73, 80, 81, 115, 117] or to AIDS [142]; women who have had an abortion [83]; those mourning the death of their pets [75, 82, 95] and; women who have experienced a traumatic birth [108].

It is interesting to note that the topics explored using SM data have become more specific over time. Earlier publications investigated grief and mourning related to death more generally. Topics have since become much more tailored to include, for example, grief related to being a mother with cancer [124], and the phenomenon of mediatized death–the intense publicizing of someone's death on SM–which many times leads to parasocial grieving–public mourning for someone not personally known [64, 65, 96, 107]. Access to data from SM appears to have facilitated a deeper understanding of areas of mourning that would not be easily researched using more traditional methods of data collection.

Natural data were used in 94% of the records. These data included text, images, audio, or video materials produced without the influence of a researcher but collected by a researcher with the intention of analyzing them in a research project. Data were researcher-generated in 4% of records. In these studies, the researcher had some influence in the generation of data, such as being a poster in an online forum where the data were collected [90] or analyzing data from a Facebook page created by the researcher [125]. The proportion of included records that sought natural data is indicative of a seismic shift in research; a shift that will no doubt continue to have impact as SM platforms are used to mine data that documents the human experience.

A wide range of approaches were used to research mourning using SM, with content analysis being the most prevalent among the qualitative records, and sentiment analysis among the quantitative records. Many records described details of the process of data collection and analysis; however, most did not explicitly report a methodology. For example, Doveling [46] described how the data were collected, and DeGroot [107] described how the coding of Facebook data was undertaken. Liu [116] described how a web-crawler was designed and deployed to download information from the selected SM platform, the process of data selection and collection, as well as the methodology for text analysis. While a comprehensive description of the process of data collection and analysis would increase transparency and academic rigor, there is the need to protect SM users' privacy, especially when details of threads and hashtags are published with direct quotes from users. Bruns [143] discusses academic scholarship in the analysis of large data sets collected online, highlighting the need for full documentation of methods in this emerging area of SM research where methods and tools are frequently being adapted and created to suit the context of online research.

One of the discrepancies noted in the included records relates to ethics, and specifically, whether review by and Institutional Review Board (IRB) and consent from SM users were sought. Two significant questions arise when considering ethics in social media research: 'Are researchers handling primary data from human subjects or can the data collected from SM platforms be considered secondary data?', and 'Are these data public or private?' There is currently no consensus to these questions and answers will ultimately guide the requirements for ethical research practice and determine what measures are needed to protect privacy and anonymity in SM research going forward.

The involvement of human participants in research has traditionally been the criteria to determine whether a project needs ethics approval from an IRB [30]. However, if researchers consider the data collected from SM to be publicly available secondary data–data previously collected (in a SM platform) for a purpose other than the current purpose (analysis for research)–this would traditionally justify an exemption from an IRB [144].

Researchers with a background related to biomedical sciences are very familiar with the ethical principles guiding human research outlined in the Belmont Report [145] namely: respect for persons, beneficence, and justice, and therefore likely default to considering SM research to involve human subjects. Whereas researchers from a background in social science disciplines may consider SM research to involve only secondary data and therefore not seek review

from an IRB. In this review the difference between reporting application to an IRB in health sciences and social sciences was significant, with 60% of records from health sciences reporting ethics approval, compared to only 17% in social sciences.

Prior to the internet, and specifically SM, the use of secondary data in research posed little risk to the person whose information was used, as it would be impossible to connect a quote to a person if correct data management strategies had been used, such as anonymizing data sets. But as researchers increasingly use SM data and publish quotes in their papers, the possibility of re-identification of online data leading back to the poster becomes an ethical concern [146]. This concern escalates where content from vulnerable people is being used in research without their consent [31].

Over the last decade, as well as an increase in the overall volume of publications using SM data to explore the experience of mourning, there has been an increase in applications for ethical review to IRBs, particularly after 2012. In 2012 the Association of Internet Researchers (AoIR) published the updated version of the guidelines for ethical conduct of research, in which it was recommended that researchers consider the risk of harm to participants when conducting research using online data [30]. Whether the publication of the AoIR guidelines led authors to consider the ethical aspects of using secondary data from SM platforms, or if the increase in IRB applications reflects a broader recognition of the risks related to SM data is hard to determine. Nonetheless, appropriate guidelines that reflect the changing landscape of electronic data availability are required.

The AoIR 2020 guidelines state that while all research conducted using data from SM must employ strategies to protect users' privacy, the responsibility of the research community is greater when research involves vulnerable people such as minors, minorities, and, among others mentioned in the guidelines: those who are grieving. Anonymity online cannot be guaranteed, and depending on how much information is available, it is possible to identify the original poster from any given post (Gerrard 2020). If a direct quote is published, it can be tracked back to the poster, exposing them to harm. In research involving sensitive topics such as abortion [83] or self-harm [23], risking the exposure of the identity of the original poster may bring significant personal risk to individuals. For this reason, the NHMRC recommends that even if research is being conducted using secondary data, informed consent and protection of participants are necessary considerations.

The 2007 Australian Code for the Responsible Conduct of Research [147] and the 2015 update of the National Statement [148] did not explicitly identify ethical considerations regarding the use of secondary data. However, the current guidelines, published in 2018, state that consent and respect for privacy should be considered in research using secondary data, specifying that even though consent may be impracticable in this context, the risk associated with the use and publication of secondary data needs to be considered. The recommendation is that consent from posters is gained, or the absence of informed consent is sanctioned by an IRB.

The 2018 update also addressed the issue of the expectation of privacy online. Are the data private or public? The guidelines differentiate the information based on intent or expectation, stating that information available online ranges from what is fully public, such as books or newspapers, to information that, while it is available publicly–such as SM platforms–'the individuals who have made it public may consider it to be private, to information that is fully private in character'[p36]. Some information is clearly public domain, while others belong to the private domain. But what if information is publicly available, but the poster intended it for a specific audience? An example of this would be research using written posts from a publicly available online forum for parents who experienced perinatal loss [138]. In this case, even

though consent was not sought from posters, raw secondary data was used, and quotes were published verbatim as data was considered public.

While exploring SM users' views on ethical conduct in SM research, researchers found that 80% of Twitter users expected to be asked prior to a researcher using content produced by them in research, and approximately 90% expected their anonymity to be protected by researches [149]. These results show a clear discrepancy in how researchers may consider SM data–as public secondary data–and users consider their content–as private expression. The NHMRC guidelines state that when the access and use of information by a researcher does not match the expectation of individuals for the use of said information, privacy concerns should be raised [150].

It is important to note that this inclusion of ethical guidelines around secondary data into a guideline of ethical conduct of human research is very recent. Many records included in this review reported their secondary data as publicly available and therefore ethics approval or consent to use data were unnecessary. As researchers understand more about the use of data from social media in Mourning 2.0, the risks associated with the re-identification of posts, as well as respect for intent and expectations of posters, strategies, protocols, and standards will be required to protect SM users.

## Implications for future research

This review has highlighted the changing landscape of research relating to grief and mourning resulting from the burgeoning use of SM. It was not the purpose or intention of the review to recommend changes in policies or practice relating to research using data from SM, but rather to map how this unique area of research has been developing over the last 20 years. Findings of this review may be useful for those wanting to undertake research that investigates grief and mourning using SM data and in the ongoing review and development of frameworks and policies that seek to provide protection to participants and strive to ensure transparency in research conduct.

## Limitations

SM terminology has changed considerably since inception, resulting in significant variation in indexing terms. Consequently, despite a comprehensive search of the literature, it is possible that not all records that met inclusion criteria were captured in this review. No critical appraisal was performed in this review in line with the recommendations underpinning a Scoping Review methodology and only records in English and Portuguese were included. Although the data extraction was piloted with the three reviewers, and quality checks were undertaken, most of the data extraction was conducted by the lead reviewer.

## Conclusions

This Scoping Review has provided insight into how SM data are used to research the experience of mourning. Through the analysis of eighty-nine records, this review has addressed four questions:

a) 'Which topics related to mourning are being studied using SM data?'; b) 'What study designs have been employed in the analysis of SM data about the experience of mourning?'; c) 'What type of data (natural or generated) have been predominantly used in SM research about the experience of mourning?' and d) 'How are ethical aspects considered in the published research?'.

The findings of this review highlighted the diversity of topics investigated using SM data, which range from the death of a loved one to grief related to life experiences. There was

significant variability in approaches to data analysis, with most records using natural data and employing qualitative approaches to analyse said data, particularly content analysis. This variability likely reflects the novelty of this approach to data collection, and consequently how researchers are experimenting with different methods of data analysis. What has become evident in this review is that, even though most records did not obtain ethics approval, researchers' perceptions of the ethical implications intrinsic to SM research have evolved over the last 10 years and will likely continue to do so as more is understood about the complexities involved in the use of secondary data from SM platforms, in research about mourning, and in other vulnerable populations. This is an emerging and rapidly changing field of research, and as such offers new opportunities to explore the social phenomenon of mourning.

## Supporting information

**S1 Appendix. Preferred Reporting Items for Systematic reviews and Meta-Analyses extension for Scoping Reviews (PRISMA-ScR) checklist.**
(DOCX)

**S2 Appendix. Search strategy for CINAHL.**
(DOCX)

**S3 Appendix.**
(XLSX)

## Acknowledgments

We would like to thank Vikki Langton, Research Librarian, for her assistance with the search strategy.

We also thank Dr. Vitor Parola, Adjunct Professor at The University Fernando Pessoa, Porto, Portugal, for his assistance with full text screening of records in Portuguese.

## Author Contributions

**Conceptualization:** Julia Muller Spiti, Paul McLiesh, Janet Kelly.

**Data curation:** Julia Muller Spiti, Paul McLiesh, Janet Kelly.

**Formal analysis:** Julia Muller Spiti, Ellen Davies, Paul McLiesh, Janet Kelly.

**Investigation:** Julia Muller Spiti, Ellen Davies.

**Methodology:** Julia Muller Spiti, Paul McLiesh, Janet Kelly.

**Project administration:** Julia Muller Spiti.

**Resources:** Julia Muller Spiti.

**Supervision:** Ellen Davies, Paul McLiesh, Janet Kelly.

**Writing – original draft:** Julia Muller Spiti, Ellen Davies.

**Writing – review & editing:** Julia Muller Spiti, Ellen Davies, Paul McLiesh, Janet Kelly.

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
