## [Decision Letter · Decision Letter 0]

29 Mar 2022

PONE-D-22-04135How social media data are being used to research the experience of mourning: A scoping reviewPLOS ONE

Dear Dr. Spiti,

Thank you for submitting your manuscript to PLOS ONE. After careful consideration, we feel that it has merit but does not fully meet PLOS ONE’s publication criteria as it currently stands. Therefore, we invite you to submit a revised version of the manuscript that addresses the points raised during the review process. Based on the review results, a major revision has been suggested in the paper before final decision. Please respond to the reviewers' comments and revise your manuscript accordingly. 

We look forward to receiving your revised manuscript.

Kind regards,

Haoran Xie

Academic Editor

PLOS ONE

Journal Requirements:

Reviewers' comments:

Reviewer's Responses to Questions

**Comments to the Author**

1. Is the manuscript technically sound, and do the data support the conclusions?

Reviewer #1: Yes

Reviewer #2: Partly

2. Has the statistical analysis been performed appropriately and rigorously? 

Reviewer #1: N/A

Reviewer #2: Yes

3. Have the authors made all data underlying the findings in their manuscript fully available?

Reviewer #1: Yes

Reviewer #2: No

4. Is the manuscript presented in an intelligible fashion and written in standard English?

Reviewer #1: Yes

Reviewer #2: Yes

5. Review Comments to the Author

Reviewer #1: 1. Introduction - Social Media and research (page 4): After reading this section, its title seemed a little bit vague and not specific. Also, the authors put diverse content all together in this part, which I thought may lead to some confusion for readers.

2. Introduction - Social Media and research (page 4): In recent years, there have been a variety of social media platforms launched with quite distinct features, such as Instagram best for sharing photos but not lengthy articles. In addition, different platforms may have dissimilar users of demographics and audience targeted, which I suggest bringing some related background/discussions.

3. Introduction - Types of SM data used by researchers (page 6): The part from the lines 86 to 96 “While research using SM data can be valuable in the quest to …” seemed to be sort of not matched to the section’s title. Might consider separating it from the section or strengthening the relevance.

4. Introduction: Ethical consideration of social media seems to be one of the topics of interest the authors want to highlight and discuss; suggest having an independent section of it in Introduction.

5. Methodology (page 8): Because some readers may not be so familiar with the scoping review methodology, it would be greatly helpful if there is still a part ahead briefly describing the whole procedure.

6. Methodology - Eligibility criteria (page 9, line 163): Here I suggest including/listing at least one or two criteria examples.

7. Methodology - Eligibility criteria (page 9, line 167): The statement “... that had expressed grief on SM” may be quite vague and not concrete to readers; I believe adding some instances that can help make it more concrete.

8. Methodology - Eligibility criteria (page 10, line 184): Why used MEDLINE and CINAHL as the initial search databases? Should provide the considerations.

9. Methodology - Eligibility criteria (page 11, line 213): Could the authors clarify what “quality control” actually means here? And what were the processes to maintain quality in detail?

10. Discussion: May have a part highlighting the practical implications for future studies, as well as policy recommendations.

11. Discussion: If can further improve the connection between paragraphs, this will make readers read the article smoother and easier.

Reviewer #2: The authors provide a short review on how SM data are being used to study mourning experience. Although the scope of this review is narrow comparing to most review articles, the topic itself is interesting and in frontier. In principle, it has the potential to be accepted, however, the submission has remarkable drawbacks and thus cannot be accepted in the current form. Specific comments and suggestions are as follows.

[1] This review provides rich materials as well as the concerned issues in related studies, while it lacks description of main findings in the mentioned works. For example, what are the answers to the main question and the four sub-questions.

[2] In the review article [J. Gao, Computational Socioeconomics, Physics Reports 2019], the authors introduced how to use SM data to reveal and predict socioeconomic phenomena. In particular, the subsection “Online posts for disease surveillance” described very similar issues to what introduced in this submission. The authors should highlight this review, as well as some closely related but missed works (mainly in the above-mentioned subsection).

[3] The introduced methodology on SM data is similar to those summarized in the Perspective [T. Zhou, Representative methods of computational socioeconomics, J. Phys. Complexity 2021], in particular, I suggest the authors use “natural data” instead of “naturally occurring data” (how to deal with natural data is one main topic in the above-mentioned Perspective). This review spent a lot on ethical issues, which are highly valuable. In addition to the current discussion, I suggest the authors pay more attention to the private protection problem. Because when natural data is used, the involved individuals are not aware of the fact that they are under investigation. Even if the data is collected from public websites, the private information has to be protected because individuals probably do not want other people knowing the analytic results related to them, or do not want to see the results themselves. For example, a user is willing to share his information to Facebook friends, which does not imply that he agrees to be known as a predicted depression patient or a predicted gay based on his shared data, see for example [M. De Choudhury, et al., AAAI Conf. Web and Social Media 2013] and [M. Kosinski, et al., Private traits and attributes are predictable from digital records of human behavior, PNAS 2013].

[4] Related to Comment [3], the authors should mention an ethical issue that one can use AI or data mining technique to deanonymized data, see for example, the two famous paper by Y.-A. De Montjoye, et al., [Unique in the crowd, Scientific Reports 2013] and [Unique in the shopping mall, Science 2015].

[5]For the “researcher generated data” (it is better to use generated data instead), if the researchers intend to interact with subjects (aware or not aware, personally or by some pre-designed SM-robots), researchers have to carefully evaluate in advance whether the materials shown to subjects, the interactions themselves, and the feedback from researchers will result in long-standing negative impacts on subjects’ emotion and mentality.

[6] In a recent work [C. Liu, et al., Emoji use in China: popularity patterns and changes due to COVID-19, Applied Intelligence 2022], emoji usage in SM was analyzed to reveal temporal patterns of sentiments in the outbreak of COVID-19, which is closely related to the COVID-19 issue mentioned in the submission. By the way, the authors should introduce the main findings of Refs. [56] and [99].

6. PLOS authors have the option to publish the peer review history of their article (what does this mean?). If published, this will include your full peer review and any attached files.

Reviewer #1: No

Reviewer #2: **Yes: **Tao Zhou

---

## [Author Response · Author response to Decision Letter 0]

20 Apr 2022

A rebuttal letter has been submitted titled: Response to Reviewers.

---

## [Decision Letter · Decision Letter 1]

23 Jun 2022

How social media data are being used to research the experience of mourning: A scoping review

PONE-D-22-04135R1

Dear Dr. Spiti,

We’re pleased to inform you that your manuscript has been judged scientifically suitable for publication and will be formally accepted for publication once it meets all outstanding technical requirements.

Kind regards,

Haoran Xie

Academic Editor

PLOS ONE

Additional Editor Comments (optional):

Reviewers' comments:

Reviewer's Responses to Questions

**Comments to the Author**

1. If the authors have adequately addressed your comments raised in a previous round of review and you feel that this manuscript is now acceptable for publication, you may indicate that here to bypass the “Comments to the Author” section, enter your conflict of interest statement in the “Confidential to Editor” section, and submit your "Accept" recommendation.

Reviewer #1: All comments have been addressed

Reviewer #2: All comments have been addressed

2. Is the manuscript technically sound, and do the data support the conclusions?

Reviewer #1: Yes

Reviewer #2: Yes

3. Has the statistical analysis been performed appropriately and rigorously? 

Reviewer #1: N/A

Reviewer #2: N/A

4. Have the authors made all data underlying the findings in their manuscript fully available?

Reviewer #1: Yes

Reviewer #2: No

5. Is the manuscript presented in an intelligible fashion and written in standard English?

Reviewer #1: Yes

Reviewer #2: Yes

6. Review Comments to the Author

Reviewer #1: This is interesting topic and the authors have already carefully dealt with all my concerns. I have no more comments.

Reviewer #2: Although there is still space to further improve, I think the current form can be accepted for publication.

7. PLOS authors have the option to publish the peer review history of their article (what does this mean?). If published, this will include your full peer review and any attached files.

Reviewer #1: No

Reviewer #2: **Yes: **Tao Zhou

---

## [Editor Report · Acceptance letter]

27 Jun 2022

PONE-D-22-04135R1 

How social media data are being used to research the experience of mourning: A scoping review 

Dear Dr. Spiti:

I'm pleased to inform you that your manuscript has been deemed suitable for publication in PLOS ONE. Congratulations! Your manuscript is now with our production department. 

Kind regards, 

on behalf of

Professor Haoran Xie 

Academic Editor

PLOS ONE